# Training Hard-Threshold Networks with Combinatorial Search in a Discrete Target Propagation Setting

## Abstract

Learning deep neural networks with hard-threshold activation has recently become an important problem due to the proliferation of resource-constrained computing devices. In order to circumvent the inability to train with backpropagation in the present of hard-threshold activations, Friesen & Domingos (2018) introduced a discrete target propagation framework for training hard-threshold networks in a layer-by-layer fashion. Rather than using a gradient-based target heuristic, we explore the use of search methods for solving the target setting problem. Building on both traditional combinatorial optimization algorithms and gradient-based techniques, we develop a novel search algorithm Guided Random Local Search (GRLS). We demonstrate the effectiveness of our algorithm in training small networks on several datasets and evaluate our target-setting algorithm compared to simpler search methods and gradient-based techniques. Our results indicate that combinatorial optimization is a viable method for training hard-threshold networks that may have the potential to eventually surpass gradient-based methods in many settings.

## 1 Introduction

Interest in network quantization has rapidly increased in recent years (Shan et al., 2018; Hubara et al., 2016; Friesen & Domingos, 2018). Network quantization reduces the complexity of a neural network by lowering the precision requirements of the activations and/or weights (Wu et al., 2018; Choi et al., 2016). As more network-based models are being deployed to mobile devices, computational power reduction and efficiency increase become critical issues. In order to train a deep neural network with full precision parameters efficiently, we depend upon powerful computational devices such as high-performance CPUs and GPUs. These devices use a lot of energy and have abundant memory, which mobile devices do not have. Learning networks with hard-threshold activations allows binary or low-precision inference and training which reduces computation time and energy demands. This seemingly innocuous change leads to a surprising challenge—networks with hard-threshold activations cannot be trained using the method of backpropagation since gradient descent requires a differentiable activation function. Unfortunately, the derivative of a hard-threshold function is 0 almost everywhere. The standard remedy is to use a gradient estimator known as the straight-through estimator (STE), first proposed by Hinton et al. (2012), and later analyzed by Bengio et al. (2013).

Friesen & Domingos (2018) designed a novel discrete target propagation framework based on the idea of separating the network into a series of perception problems each with artificially set targets. They gave a simple gradient-based target setting heuristic using the sign of the next layer's loss gradient. Instead, we propose the use of search methods for target setting. The application of combinatorial optimization methods to training neural networks remains largely unexplored. Discrete target propagation frameworks offer an opportunity to apply these methods in deep learning. We initiate a feasibility study on search methods for learning hard-threshold networks. This may open the door to significantly improving performance by leveraging state-of-the-art combinatorial optimization algorithms.

Building on the convex-combinatorial optimization framework of Friesen & Domingos (2018), we present a search algorithm for target setting. We also develop a novel gradient-search hybrid target setting algorithm. We demonstrate the effectiveness of our method in training small networks on several data sets, with performance approaching the state of the art achieved by Friesen & Domingos (2018) for hard-threshold networks. We evaluate all target-setting algorithms mentioned in this paper, indicating significant room for improvements.

## 1.1 RELATED WORK

Using the conventional backpropagation algorithm to train a neural network via gradient descent, the gradient of the loss is computed with respect to the output of each layer. This is propagated recursively from the output layer to the input layer using the chain rule. In our case, the activation function at each hidden layer would involve a non-linearity, meaning that the partial derivative will be zero, hence ruling out the use of backpropagation. In the case that we cannot use gradient descent, we use a gradient estimator. The straight-through estimator (STE) (Hinton et al., 2012; Bengio et al., 2013) backpropagates through the hard threshold function as if it were the identity function. When considering a single layer of neurons, the STE has the right sign, but once it is used through more hidden layers, this is not guaranteed.

Our work builds on the idea of target propagation, first proposed by LeCun et al. (1989). The main idea is to compute targets rather than gradients, at each layer. Since then many different methods of target propagation have been proposed (Bengio & Frasconi, 1995; Courbariaux et al., 2015). In particular, Bengio (2014) proposed a method called "*vanilla target propagation*" to avoid the chain of derivatives through the network by considering an "approximate inverse". Later, Hubara et al. (2016) improved this method, claiming that the imperfection of the inverse yields severe optimization problems. This led to their *difference target propagation*, which is a linearly corrected formula of Bengio (2014). They showed that this method of target propagation performs comparable to backpropagation methods.

Most recently, Friesen & Domingos (2018) show that learning a deep hard-threshold network reduces to finding a feasible setting of its targets and then optimizing its weights in a layer-local fashion. Peng et al. (2018) introduce a new technique which they call SPIGOT: structured projection of intermediate gradients optimization technique. SPIGOT is a backpropagation technique for neural networks with hard-threshold predictions. It defines gradient-like quantities associated with intermediate non-differentiable operations, allowing backpropagation before and after them.

## 2 FEASIBLE TARGET PROPAGATION

We begin by describing *feasible target propagation*, the method introduced by Friesen & Domingos (2018) for training deep hard-threshold networks upon which our work is based. Consider a dataset $\mathcal{D} = (X, T)$ composed of an $n \times m$ real-valued matrix $X$ of vector-valued inputs $\{\mathbf{x}^{(j)}\}_{j=1}^{m}$ and a corresponding $\{+1, -1\}$-valued vector $T$ of targets $\{t^{(j)}\}_{j=1}^{m}$. The aim is to learn an $\ell$-layer neural network with hard-threshold activations

$$f(x; W) = g(W_\ell g(W_{\ell-1} \cdots g(W_2 g(W_1 x)))),$$

with input vector $x \in \mathbb{R}^n$, weight matrices $W = \{W_i \mid W_i \in \mathbb{R}^{n_i \times n_{i-1}}, 1 \leqslant i \leqslant \ell\}$, and activation function $g(x) = \text{sign}(x)$, where $n_0 = n$ and $\text{sign}(x) = 1$ if $x > 0$ and $-1$ otherwise. For simplicity, the bias terms are incorporated into the weight matrices. Let $Z_i = W_i g(W_{i-1} \cdots g(W_1 X))$ and $H_i = g(Z_i)$ denote the full-batch pre-activation and post-activation matrices, and $\mathbf{z}_i = (z_{i1}, \ldots, z_{in_i})$ and $\mathbf{h}_i = (h_{i1}, \ldots, h_{in_i})$ denote the corresponding per-instance vectors. The learning goal is to find weights $W$ of $f$ that minimize the training loss $L(H_\ell, T) = \sum_{j=1}^{m} L(\mathbf{h}_\ell^{(j)}, t^{(j)})$ for some per-instance loss $L(\mathbf{h}_\ell, t)$.

Because the derivative of the sign function is zero almost everywhere, backpropagation cannot be used to train the network $f$ when $\ell > 1$. Instead, Friesen & Domingos (2018) considered the decomposition $f_\ell \circ f_{\ell-1} \circ \cdots \circ f_1$ of $f$ into perceptrons $f_i(\mathbf{h}_i) = \mathbf{h}_{i+1} = g(W_i \mathbf{h}_i)$. Given a matrix $T_i$ of targets and a loss function $L_i$ for each perceptron $f_i(H_i)$, we can use the perceptron algorithm, or a gradient descent method, to set $W_i$ such that $f_i$ produces these targets (if the dataset $(H_i, T_i)$ is

linearly separable) or minimizes the loss $L_i(Z_i, T_i)$. The problem then becomes setting the targets $T = \{T_1, \ldots, T_\ell\}$ in such a way as to minimize the output loss $L(H_\ell, T)$.

Towards that end, Friesen & Domingos (2018) proposed a recursive approach that involves setting the targets $T_i$ of $f_i$ in order to minimize the next layer's loss $L_{i+1}(Z_{i+1}, T_{i+1})$. More specifically, the method proceeds as follows. For each minibatch $(X_b, T_b)$ from dataset $\mathcal{D}$, initialize $T_\ell = T_b$ and $T_1, \ldots, T_{\ell-1}$ as the activation of the corresponding layer in $f(X_b; W)$ and, starting with $i = \ell$, for all $1 \leqslant i \leqslant \ell$,

1. assign the targets $\hat{T}_{i-1}$ for the next layer based on the current weights $W_i$ and loss $L_i(f_i(T_{i-1}), T_i)$,
2. update $W_i$ with respect to the loss $L_i(f(T_{i-1}), T_i)$, and
3. set $T_{i-1} \leftarrow \hat{T}_{i-1}$.

## 2.1 RELATIONSHIP TO GRADIENT ESTIMATION

In their work, Friesen & Domingos (2018) used the heuristic

$$t_{ij} = \text{sign}(-\frac{\partial}{\partial h_{ij}} L_{i+1}(Z_{i+1}, T_{i+1})) \tag{1}$$

for setting the $j$th component $t_{ij}$ of the target vector $\mathbf{t}_i$ at layer $i$, and the layer loss

$$L_i(z_{ij}, t_{ij}) = (\tanh(-t_{ij} z_{ij}) + 1) \left| \frac{\partial L_{i+1}}{\partial h_{ij}} \right|, \tag{2}$$

where $\tanh(-tz) + 1$ is the *soft hinge loss*, so called because it is a smooth approximation of the saturated hinge loss $\max(0, 1 - \max(tz, -1))$ commonly used for training classifier networks. This combination of target heuristic and layer loss function allows information to be transmitted from the output loss back through the network to every layer, enabling effective learning as empirically demonstrated by Friesen & Domingos (2018).

They briefly commented on the relationship between this form of feasible target propagation and the straight-through estimator, but we believe it is worth further consideration here. The gradient of the weighted loss $L_i$ given in (2) with respect to the pre-activation $z_{ij}$ is

$$\frac{\partial L_i(z_{ij}, t_{ij})}{\partial z_{ij}} = -\text{sign}\left(-\frac{\partial L_{i+1}}{\partial h_{ij}}\right) \text{sech}^2\left(-\text{sign}\left(-\frac{\partial L_{i+1}}{\partial h_{ij}}\right) z_{ij}\right) \left| \frac{\partial L_{i+1}}{\partial h_{ij}} \right|$$

$$= \text{sech}^2(z_{ij}) \frac{\partial L_{i+1}}{\partial h_{ij}}$$

since $\text{sech}$ is an even function. This implies that the method of Friesen & Domingos (2018) is equivalent to a $\text{sech}^2$ gradient estimator, that is, the straight-through estimator with the identity function replaced with $\text{sech}^2$.

This is significant for two reasons. First, it suggests a simpler, independent justification for the performance improvements obtained by their method. Gradient estimation can be viewed as a way of propagating errors from the output layer by pretending the network has a different activation function. In particular, the saturated straight-through estimator corresponds to using a truncated identity function network as the error propagation proxy, while the $\text{sech}^2$ gradient estimator corresponds to instead using a $\tanh$ network. When the activation function of the proxy network approximates the hard-threshold activation (in our case, the sign function) of the actual network (see Figure 1), we may expect the network to train effectively. From this perspective, the contributions to the state of the art by Friesen & Domingos (2018) reduce to finding an approximation to the sign function, the hyperbolic tangent, with more favorable properties for training; specifically, the derivative of $\tanh$ is nonzero outside the range $[-1, 1]$, unlike the derivative of the truncated identity function. Although this justification and, moreover, the relationship between the straight-through estimator and their method were noted by Friesen & Domingos (2018), recognizing that this is the sole contributor to the obtained performance improvements clarifies the role of their method. It also leads into our second point, which is that this line of reasoning motivates investigating alternative implementations of the discrete target propagation framework, which may not bear any connection to gradient estimation.

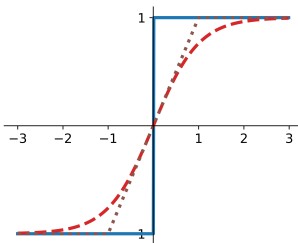

Figure 1: The sign activation (solid blue line) and hyperbolic tangent (dashed red line) and truncated identity (dotted brown line) activations, which approximate the sign function while being differentiable on a set of positive measure.

## 3 COMBINATORIAL SEARCH FOR TARGET SETTING

We propose an alternative to the target setting heuristic (1) used in Friesen & Domingos (2018). The target setting problem at layer $i$ can be succinctly written as the following optimization problem

$$\min \ L_i(f_i(T_{i-1}), T_i) \quad \text{s.t. } T_{i-1,j} \in \{-1, 1\} \ \forall j \tag{3}$$

wherein the the subsequent candidate targets at layer $T_i$ are treated as data. The feasible region is the vertices of a hypercube, and in general the objective function is non-convex. This justifies the use of heuristics, such as local search, for this problem.

### 3.1 RANDOM LOCAL SEARCH

The procedure of *random local search* is highly flexible and many variations are possible. Over the course of training a neural network via feasible target propagation we have to solve very many large instances of problem (3). With that in mind we elected to use a very lightweight variation of random local search, presented in Algorithm 1.

#### 3.1.1 IMPROVEMENTS OVER NAIVE LOCAL SEARCH

Our full method involves two improvements over the naive approach to local search. We improve 1) how the starting target is generated, and 2) how the probability of exploring a neighbour is chosen.

**Generating a Seed Candidate** As in the naive approach we begin with a uniform random seed candidate $T$ then we generate a subset $\mathcal{N}_T$ of the neighbourhood of $T$ as in step $2(b)$ of our method using a uniform random probability distribution.

Now we will choose our starting candidate using gradient information from each of these random sampled candidates. Our idea is to apply in the sign function to the average of the gradient evaluated at each point in $\mathcal{N}$. In particular we choose (applying the sign function coordinate-wise) $T_i = \text{sign}(\frac{1}{|\mathcal{N}|} \sum_{C \in \mathcal{N}} \frac{\partial L_{i+1}}{\partial T_i}(C))$.

**Setting the Probabilities** Consider an entry $h$ of $T_i$. Recall that the gradient $\nabla L_{i_1}$ points in direction of local steepest increase. For intuition, imagine that $L_{i+1}$ is a convex function. Then if we were to flip the sign of entry $h$ when $\text{sign}(\frac{\partial L_{i+1}}{\partial h}) \neq \text{sign}(h)$ we would surely be increasing the value of $L_{i+1}$. This guides us to the following heuristic approach: flip entries only when $\text{sign}(\frac{\partial L_{i+1}}{\partial h}) = \text{sign}(h)$ and do so in a manner proportional to $|\frac{\partial L_{i+1}}{\partial h}|$. In particular we flip with probability $\min\{|\frac{\partial L_{i+1}}{\partial h}| + \text{offset}, 1\})$ where offset is some positive constant. Typically offset $= \frac{3}{5}$ works well in our experience, as we want a relatively high probability of following the direction indicated by the gradient.

## 4 EXPERIMENTS

To evaluate our combinatorial search-based target setting methods on training hard-threshold networks via discrete target propagation, we begin by comparing the performance of models trained

---

**Algorithm 1:** Guided Random Local Search (GRLS)

---

1 $\underline{\text{function GRLS}(i, \alpha, a, \beta, \gamma, \delta, \omega)};$

**Input** : $i$: layer of network to set targets (through which we access loss function $L$ and
dimensions for targets $T$).
$\alpha$: probability any given entry of initial random candidate is 1.
$a$: vector of probabilities for flipping sign of an entry of initial random candidate when
generating neighbours.
$\beta$: number of neighbours to average over in generating initial random seed.
$\gamma$: number of rounds of local search.
$\delta$: number of random neighbours to check during local search.
$\omega$: additive offset to flip probability for gradient-guided random neighbour generation.

**Output:** $T_i$: targets for layer $i$.

2 Let $T$ be a candidate such for each entry of $h$ of $T$: $Pr(h = 1) = \alpha$, and $Pr(h = -1) = 1 - \alpha$.

3 $\mathcal{N}_T \leftarrow \emptyset$

4 **for** $\beta$ *iterations* **do**

5     $C \leftarrow T$

6     **for** each entry $h$ of $C$**:** flip sign of $h$ with probability $a_h$.

7     $\mathcal{N}_T \leftarrow \mathcal{N}_T \cup \{C\}$

8 **end**

9 $T_i \leftarrow \text{sign}\left(\frac{1}{|\mathcal{N}_T|} \sum_{C \in \mathcal{N}_T} \frac{\partial L_{i+1}}{\partial T_i}(C)\right)$

10 **for** $\gamma$ *rounds* **do**

11     $\mathcal{N} \leftarrow \emptyset$

12     **for** $\delta$ *iterations* **do**

13         $C \leftarrow T_i$

14         **for** *each entry $h$ of $C$* **do**

15             **if** $\text{sign}\left(\frac{\partial L_{i+1}}{\partial h}\right) \neq \text{sign}(h)$ **then** flip sign of $h$ with probability $a_h p_h$, where
$p_h := \min\{|\frac{\partial L_{i+1}}{\partial h}| + \omega, 1\}$

16         **end**

17         $\mathcal{N} \leftarrow \mathcal{N} \cup \{C\}$

18     **end**

19     $T_i \leftarrow \arg\min\{L_i(f_i(C), T_{i+1}) : C \in \mathcal{N}\}$

20 **end**

21 **return** $T_i$

---

with our methods to models trained using gradient estimation, including the standard saturated
straight-through estimator and the state-of-the-art $\tanh$ estimator of Friesen & Domingos (2018).
We trained several different networks on three standard image classification datasets, MNIST (Le-
Cun & Cortes (1998)), CIFAR-10 and CIFAR-100 (Krizhevsky (2009)), as well as a graph classi-
fication dataset. On MNIST, we trained a small 2-layer feedforward network and a simple 4-layer
convolutional network. On CIFAR-10 and CIFAR-100, we trained the same 4-layer convolutional
network, a simpler version of the 8-layer convolutional network of Zhou et al. (2016). This is the
same convolutional network used by Friesen & Domingos (2018), allowing for direct comparison
with their results.

For the purpose of testing our methods in a highly constrained setting, we assembled a small graph
connectivity classification dataset. The dataset contains all graphs on six vertices labeled by their
connectivity. There are 32768 graphs in total, 26704 connected graphs and 6064 disconnected
graphs, and we randomly partitioned the dataset into training and test sets containing 29491 and
3277 graphs, respectively. We chose this dataset for its connection to discrete optimization, the
basis of our methods, and with the inclination that the discrete nature of graph connectivity might
demonstrate a potential advantage of our combinatorial search methods over gradient estimation
in certain settings. On this dataset, denoted GRAPHS, we trained the small 2-layer feedforward
network.

Final test accuracies for the 2-layer feedforward network and 4-layer convolutional network on
each of the datasets are shown in Table 1. GRLS achieves a significantly higher test accuracy on

|  | GRLS | SSTE | FTPROP |
|---|---|---|---|
| 2-layer feedforward net (GRAPHS) | **91.9** | 79.1 | 88.3 |
| 2-layer feedforward net (MNIST) | 96.2 | 95.6 | **96.7** |
| 4-layer convnet (MNIST) | **98.5** | 98.1 | **98.5** |
| 4-layer convnet (CIFAR-10) | 70.7 | (80.6) | **80.6** |
| 4-layer convnet (CIFAR-100) | 46.3 |  | 49.5 |

Table 1: Final test accuracies obtained after training each network for 150, 300, or 200 epochs (GRAPHS/MNIST, CIFAR-10, and CIFAR-100, respectively) on the corresponding dataset.

GRAPHS than the SSTE and FTPROP, by a margin of 12.8% and 3.6% respectively; given that this is the only dataset on which it outperforms both FTPROP and SSTE, this suggests that our prior speculation may be correct and a highly discrete task such as graph connectivity classification is likely less amenable to gradient-based target setting methods and more suited to search-based techniques, potentially due to reduced smoothness in the loss landscape. As the network grows in size and the datasets become more complex, GRLS tends decrease in performance faster than the SSTE and FTPROP. While it is able to roughly match the performance of FTPROP on MNIST with either network, and exceed the accuracy of the SSTE on MNIST for both networks by 0.6% and 0.4%, respectively, the gap between GRLS and FTPROP widens by 9.9% on CIFAR-10. This begins to indicate that the higher dimensionality of the CIFAR-10 data manifold compared to MNIST may play a much larger role in inhibiting the performance of GRLS than increasing the size of the target search space. In fact, relative to FTPROP, GRLS achieves a 0.5% higher accuracy on MNIST with the 4-layer convolutional network than with the 2-layer feedforward network, suggesting that the convolutional layers ease the search problem or make it more suited to the specific search strategies of GRLS.

|  | 2-layer FFN | | 4-layer CNN | |
|---|---|---|---|---|
|  | GRAPHS | MNIST | MNIST | CIFAR-10 |
| SSTE | 81.3 | 95.6 | 99.2 |  |
| −LW | **89.3** | XX.X | **99.4** |  |
| FTPROP | **91.5** | **96.7** | 99.4 | **93.5** |
| −LW | 88.4 | 92.4 | 97.7 | 67.8 |
| GRLS | 93.9 | **96.5** | 99.4 | **74.0** |
| −GG | 92.5 | **96.5** | 99.4 | 70.1 |
| −GS | 90.1 | **96.4** | 99.4 | 58.2 |
| −(GG + GS) | 84.7 | 93.3 | 87.1 | 30.2 |
| −LW | **95.5** | 92.5 |  | 46.7 |
| −CW | 92.9 | **96.3** | 99.4 | 49.0 |
| −(CW + LW) | 93.7 | 92.8 |  | 26.0 |

Table 2: Validation Accuracies of Method Variations

## 4.1 METHOD VARIATIONS

We performed an ablation study measuring the validation accuracies for variations of the methods considered in this paper. For SSTE, FTPROP, and GRLS we considered the effect of dropping Loss Weighting (LW) from the methods. For GRLS we also considered removing Gradient Guiding (GG), Gradient Seeding (GS), and Criterion Weighting (CW) (weighting the target choosing criterion (loss) by the next layer's loss gradient). We ran experiments on the 2-layer feedforward network and the 4-layer convolutional network, and tested two distinct datasets on each network. The results are summarized in Table 2.

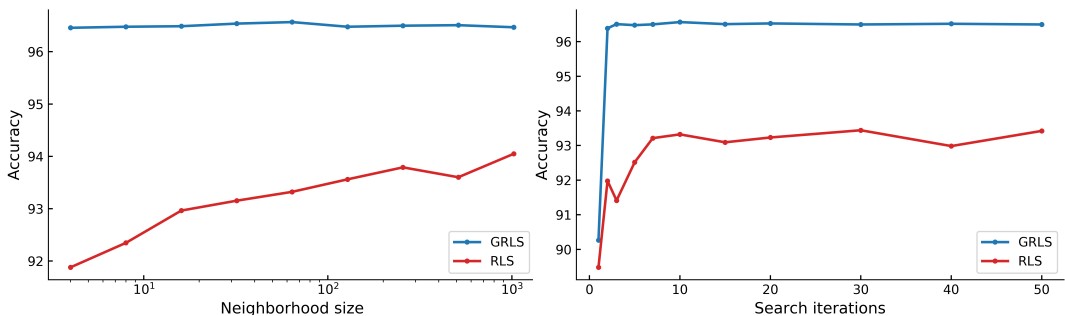

Figure 2: Effect of Varying Search Parameters on Validation Accuracy

When removing GG and GS from GRLS we lose $12.3\%$ accuracy on MNIST, and we lose a dramatic $43.8\%$ on CIFAR-10. Even further, dropping either GG or GS leads to large decrease in accuracy for GRLS on CIFAR-10. This suggests the significant role gradient guiding and gradient seeding play in randomized local search; pure RLS has only a $30.2\%$ accuracy on CIFAR-10. Another lesson from Table 2 is that Loss Gradient Weighting is necessary for good accuracy across all methods on the more complex datasets. Interestingly on the Graphs dataset, removing Loss Weighting leads to improvements for both SSTE and GRLS.

## 4.2 Sensitivity to Hyperparameters and Role of Dimensionality

There are two important hyperparameters to consider varying when studying GRLS: the number of iterations $\gamma$, and the number of candidates $\delta$. Increasing these parameters leads to increasing the strength of local search; $\delta$ value increases the breadth, and larger $\gamma$ increases the depth. We compared the accuracy of GRLS and RLS in training our 2-layer feedforward neural network on the MNIST dataset for various values of $\gamma$ and $\delta$. In one experiment we increase $\delta$ from 2 to 1024 in powers of 2, and the other we increase $\gamma$ from 1 to 50. The results on shown in Figure 2, with the $\delta$ experiment on the left, and the $\gamma$ experiment on the right.

The most striking phenomenon the data shows is a robustness of GRLS to these hyperparameter variations. In light of the performance variations on MNIST in Table 2, it seems that the Gradient Seeding technique is primarily responsible for this robustness, in particular it starts the search in a good position where further iterations do not yield significant improvement. As expected, RLS performs better as we increase $\delta$, but curiously it seems GRLS is independent of $\delta$. We believe this suggests the target setting problem exhibits a pervasiveness of local optima, each of which are close enough to the global optimum to be used effectively for target setting.

We investigate the effect of varying the number of hidden units on our training method. We test the accuracy of GRLS for training our 2-layer feedforward network with $d$ hidden units for $d \in \{20, 50, 100, 200, 400, 800, 2000, 5000\}$, on the MNIST dataset. The results are presented in Table 3. We keep the parameters $\delta$ and $\gamma$ constant as we vary $d$. For local search to be able to find good solutions $\delta$ and $\gamma$ need to scale with the input size to the search problem, which $d$ essentially corresponds to. Hence the the Loss value for the target found by RLS will increase with $d$. A decay in accuracy would then be evidence of the relationship between target setting quality and accuracy. Indeed the data shows such a decline in RLS accuracy for large $d$. Interestingly, GRLS seems to show a similar curve to FTPROP in terms of its accuracy on this dataset, lending credibility to the power of gradient-guided techniques to scale with $d$.

|  | GRLS | RLS | FTPROP |
|---|---|---|---|
| Loss Mean | **2.446410** | 2.449184 | 2.446924 |
| Loss Standard Error | 0.005673 | 0.000709 | 0.000533 |

Table 3: Mean Loss of Target Setting Methods

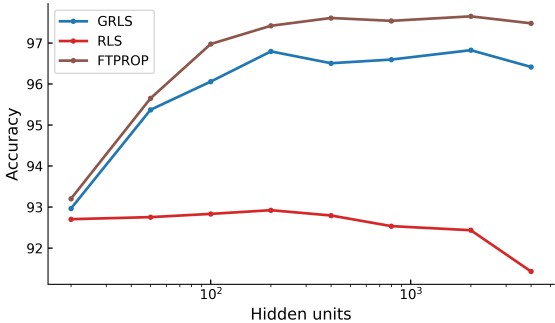

Figure 3: Effect of Varying Dimension on Validation Accuracy

### 4.3 METHOD SUBOPTIMALITY

We seek to compare the quality of target setting procedures considered in our study. We considered random instances of Problem (3) (i.e randomly generated weight matrix and output targets, with 10 possible output classes) and tasked GRLS, RLS, and FTPROP to set targets. We performed 1024 independent trials and collected the data in Table 3. We show there the loss mean and standard error of each given target setting method.

The losses for all three methods were comparable, but high as they were all over 2. Note that if a network is separable in the sense of Friesen & Domingos (2018), zero loss is possible. This indicates that there is considerable potential for improved target setting procedures that get a lower loss. Furthermore, GRLS exhibits a higher standard error than other methods, enough to potentially place the true empirical mean loss substantially lower than for FTPROP, indicating that it obtains notably better (and notably worse) targets in different trials.

## 5 FUTURE WORK

We view our work as a preliminary investigation into combinatorial techniques for target setting in training hard-threshold networks. We have shown that there is a lot of potential for such approaches, and plenty of room for improvement. An important problem to solve in future work is to develop more computationally-tractable combinatorial search algorithms. The most significant barrier for our approach is the need to perform a forward pass through the network for each candidate target considered during the search. One may alleviate this problem by instead evaluating candidates by taking their scalar product with a loss gradient or previous candidates; in the latter case, choosing candidates with positive scalar product ensures that the loss cannot decrease. In a different direction, the locality of the receptive fields of a convolutional layer could be used to enable localized target evaluation and, thereby, effective splicing together of multiple candidates to create a significantly improved target.

In keeping with the wide open potential for this line of investigation we present some open questions. Computational tools such as BARON (Tawarmalani & Sahinidis, 2005; Sahinidis, 2017) exist, which can solve discrete non-linear optimization problems, including discrete target setting, to optimality. It would be interesting to investigate the effect such a solver would have on training accuracy if used to set targets instead of local search. Generalizing from this, the combinatorial optimization literature is vast, and there are a wealth of unexplored techniques which could be tested for solving Problem (3). Further, since combinatorial training techniques may have radically different behaviour from gradient-based continuous methods in general, it would be worthwhile to study the effect of changing network architectures on training in this context. It is conceivable that different network structures from those used in the traditional best-practices can be more amenable to this approach to training. Similarly, the GRAPHS dataset seems to indicate that for some datasets local search based training can be superior to continuous methods. It would be interesting to attempt to characterize those datasets for which this phenomenon holds.

ACKNOWLEDGMENTS

We acknowledge the support of the ██████████████████████████████████████████
██████████.    We  would  like  to  thank  ████████  for  inspiring  our  interest  in  this  field,  and
████████████████  for his contributions to an early project from which this research grew.

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

## 6 APPENDIX

### 6.1 EXPERIMENT DETAILS

In all experiments involving our training method, we set the number of search iterations $\gamma = 10$, neighborhood sizes $\beta = \delta = 64$, initial random candidate probability $\alpha = 1/2$, and flip probability offset $\omega = 3/5$. Because the number of hidden units per layer varies by network, the size of the search space varies nonlinearly between different networks and distinct layers within a given network. Consequently, we vary the perturbation probability $a$ by network and layer; in particular, we set each $a_h = 5/32$ for the 2-layer feedforward network and, for $P = (0.0127, 0.0172, 0.031)$, set each $a_h = P_i$ for the 4-layer convolutional network. These correspond to an expectation of flipping approximately 15 candidate target entries at a time for the single hidden layer of the feedforward network, and 117, 70, and 31 entries for the three larger hidden layers of the 4-layer convolutional network.

As in the work of Friesen & Domingos (2018), we train all networks using the Adam optimizer (Kingma & Ba (2015)) (applied to each layer individually in the case of discrete target propagation) with a batch size of 64, weight decay coefficient $5 \times 10^{-4}$, and learning rate $2.5 \times 10^{-4}$—additional hyperparameter details can be found in the appendices. We used a single NVIDIA P100 GPU to train the convolutional and feedforward networks. For our search methods, at each training step we require a single forward pass through every layer for each candidate target generated during each iteration of the search, implying that the method requires approximately a factor of $\delta\gamma/3$ more computations than training a network with gradient estimation or, in the case of full-precision networks, backpropagation. This is somewhat alleviated by the parallel processing capabilities of GPUs, but does imply that training the small feedforward network and 4-layer convolutional network with our method is approximately $\times 3$ and $\times 100$ slower, respectively, than using gradient estimation. We view our work as an initial exploration of the use of combinatorial optimization methods in training artificial neural networks and leave this speed issue as a open problem for future study; note though that these issues do not effect the ability to perform fast and memory-efficient inference using the trained models, a key advantage of hard-threshold networks.

