# OpenReview forum: "Training Hard-Threshold Networks with Combinatorial Search in a Discrete Target Propagation Setting"
_ICLR.cc/2019/Conference_

### Official Review · AnonReviewer1 · 2018-11-01
**Borderline paper; interesting approach but insufficient contribution to warrant acceptance**

**Rating:** 5
**Confidence:** 5

**Review:**

Summary:
The paper presents a novel combinatorial search algorithm for the discrete target propagation framework developed in Friesen & Domingos (2018). Experiments on small datasets with small models demonstrate some potential for the proposed approach; however, scalability remains a concern.

  Pros:
-	I like the goal of the work and think that if the targeted problem were to be solved it would be an interesting contribution to the field.
-	The proposed search algorithm is reasonable and works OK.
-	The paper is mostly well written and clear.
-	The experiments are reasonably thorough.

  Cons:
-	The paper states that it is a feasibility study on search methods for learning hard-threshold networks, however, it only evaluates the feasibility of one combinatorial search method.
-	It’s not made clear whether other approaches were also investigated or what the authors learned from their exploration of this approach.
-	The actual algorithm is not very well explained, despite being the main contribution of the paper.
-	The datasets and models are small and not necessarily representative of the requirements of the field.
-	Scalability remains a serious concern with the proposed approach.
-	It’s not clear to me that the paper presents a sufficient enough contribution to warrant acceptance.

Overall, I like the direction but do not feel that the paper has contributed enough to warrant acceptance. The authors should use the experiments they’ve run and also run more experiments in order to fully analyze their method and use this analysis to improve their proposed approach.


Questions and comments:

1.	Did you try alternative local search algorithms or did you just come up with a single approach and evaluate it? What did you learn from the experiments and the development of this algorithm that will let you create a better algorithm in the next iteration?

2.	I think that it is unfair to say that “it suggests a simpler, independent justification for the performance improvements obtained by their method.” in reference to the work of Friesen & Domingos (2018), given that the straight-through estimator is not well justified to begin with and their work in fact provides a justification for it. I do agree that it is important to investigate alternative heuristics and approaches within the discrete target propagation framework, however.

3.	Sections 2 and 3 do not clearly define what L_i is and where it comes from. Since these do not normally exist in a deep network they should be clearly defined.

4.	“step 2(b)” is not well defined in section 3.1.1. I assume that this refers to lines 4-8 of Algorithm 1? The paper should explain this procedure more clearly in the text. Further, I question the locality of this method, as it seems capable of generating any possible target setting as a neighbor, with no guarantee that the generated neighbors are within any particular distance of the uniform random seed candidate. Please clarify this.

5.	I believe that a negative sign is missing in the equation for T_i in ‘Generating a seed candidate’. For example, in the case where |N| = 1, then T_i = sign(dL/dT_i) would set the targets to attain a higher loss, not lower. Further, for |N|=1, this seems to essentially reduce to the heuristic method of Friesen & Domingos (2018).

6.	In the ‘Setting the probabilities’ section:
(a) All uses of sign(h) can be rewritten as h (since h \in {-1, +1}), which would be simpler.
(b) The paper contradicts itself: it says here ‘flip entries only when sign(dL/dh) = sign(h)’ but Algorithm 1 says ‘flip entries only when sign(dL/dh) != sign(h)’. Which is it?
(c) What is the value of a_h in the pseudocode? (i.e., how is this computed in the experiments)

7.	In the experiments, the paper says that ‘[this indicates] that the higher dimensionality of the CIFAR-10 data manifold compared to MNIST may play a much larger role in inhibiting the performance of GRLS.’ How could GRLS overcome this? Also, I don’t agree with the claim made in the next sentence – there’s not enough evidence to support this claim as the extra depth of the 4-layer network may also be the contributing factor.

8.	In Table 2, why are some numbers missing? The paper should explain what this means in the caption and why it occurs. Same for the bolded numbers.

9.	The Loss Weighting, Gradient Guiding, Gradient Seeding, and Criterion Weighting conditions are not clearly defined but need to be to understand the ablation experiments. Please define these properly.

10.	The overall structure of the algorithm is not stated. Algorithm 1 shows how to compute the targets for one particular layer but how are the targets for all layers computed? What is the algorithm that uses Algorithm 1 to set the targets and then set the weights? Do you use a recursive approach as in Friesen & Domingos (2018)?

11.	In Figure 2, what dataset is this evaluation performed on? It should say in the caption. It looks like this is for MNIST, which is a dataset that GRLS performs well on. What does this figure look like for CIFAR-10? Does increasing the computation for the heuristic improve performance or is it also flat for a harder dataset? This might indicate that the initial targets computed are useful but that the local search is not helping. It would be helpful to better understand (via more experiments) why this is and use that information to develop a better heuristic.

12.	It would be interesting to see how GRLS performs on other combinatorial search tasks, to see if it is a useful approach beyond this particular problem.

13.	In the third paragraph of Section 4.2, it says ‘The results are presented in Table 3.’ I believe this should say Figure 3. Also, the ordering of Figure 3 and Table 3 should be swapped to align with the order they are discussed in the text. Finally, the caption for Table 3 is insufficiently explanatory, as are most other captions; please make these more informative.

14.	In Section 4.3:
(a), the paper refers to Friesen & Domingos (2018) indicating that zero loss is possible if the dataset is separable. However, what leads you to believe that these datasets are separable? A more accurate comparison would be the loss for a powerful non-binarized baseline network.
(b) Further, given the standard error of GRLS, it’s possible that its loss could be substantially higher than that of FTPROP as well. It would be interesting to investigate the cases where it does much better and the cases where it does much worse to see if these cases are informative for improving the method.

15.	Why is there no discussion of training time in the experiments? While it is not surprising that GRLS is significantly slower, it should not be ignored either. The existence of the Appendix should also be mentioned in the main paper with a brief mention of what information can be found in it.

16.	In Algorithm 1, line 2 is confusingly written. Also, notationally, it’s a bit odd to use h both as an element and as an index into T.

17.	There are a number of capitalization issues in the references.

18.	The Appendix refers to itself (“additional hyperparameter details can be found in the appendices”).

---

> ### Author Response · Authors · 2018-11-14
> **Thank you for the thorough review. We’ve addressed some of your points in the comment body**
>
> 1. We began with the most basic and general approach to local search for the target setting algorithm, which we present as the “naive” approach in the paper. We went through many possible improvements to the method and ultimately presented as GRLS the best alternative we could develop. In the future we’ll expand on this point in the paper, continue to improve our algorithm (especially to address scalability concerns), and compare with alternative search methods attempted along the way (e.g. a genetic algorithm, and others mentioned by Reviewer #3).
>
> 2. Perhaps it was unclear, but we did not intend to offer a justification for the straight-through estimator, so much as make it clear where exactly the performance improvements offered by their method originate. In particular, that their method is ultimately just an improved gradient estimator. As you mentioned, this serves mainly to motivate the consideration of search methods, rather than going a different direction, and, for example, further improving the gradient estimation approach (although that may still be of interest to others).
>
> 3. Agreed
>
> 4. This is a typo from an earlier iteration of the paper, will be fixed in revision. With respect to the local nature of the search: when the probability of flipping an entry is low, the expected number of entries in a target vector to be flipped is low and this induces in expectation a local neighborhood around a candidate target vector. In our experiments we set the flip probabilities to achieve an expectation of about 5% of the target vector flipped.
>
> 6. a) We will make this simplification. b) this is quite a serious typo. Algorithm 1 should read “=“, as dictated by the reasoning in the “Setting the probabilities” section.
>
> 7. We will likely need to address scalability issues of GRLS (as noted by other reviewers) before we can consider how GRLS could overcome this difficulty. With respect to the claim you mentioned, we offered that claim under the supposition that a search-based method such as GRLS would have more difficulty solving the credit assignment problem at lower layers than FTPROP as the number of layers increases, but you are right that more evidence is needed and this claim could be false.
>
> 8. Missing entries are caused by parameter-experiment combinations that were not tested, but would be included in a revised paper. Bolded entries indicate the best performing parameter set for the given method on the dataset corresponding to the column the entry lies in.
>
> 9. Indeed we will provide definitions in revisions.
>
> 10. Algorithm 1 is a proposed alternative to solve the target setting subproblem which occurs at each layer during a single pass of Friesen and Domingos target propagation method.
>
> 11. This was a chart for MNIST. We will make that clear. With respect to providing the chart for CIFAR10, our efforts were severely compute-constrained and it would have taken us about 3-4 weeks to generate a comparable chart for CIFAR10. Surely it is something we can include in the future though.
>
> 12. Quite possibly. It is outside the scope of this particular paper, but we are actively thinking about other applications.
>
> 13. Yes, we will address these.
>
> 14. a)The datasets are likely not separable. The claim was more to indicate that 0 is the best analytical lower bound we have on the loss. Indeed the loss on a non-binarized network would provide a better bound.
> b) Yes, we certainly hope to do that in the future.
>
> 15. In the appendix we discuss the additional costs to training time of using GRLS. Perhaps we could afford to specifically measure these during experiments though. We will make reference to what is contained in the appendices in the main body.
>
> 18. The quoted statement is not false --- but yes, that was a mistake and we will delete that reference.

---

> > ### Comment · AnonReviewer1 · 2018-11-16
> > **Thank you for the response**
> >
> > I look forward to the next version of this work.

---

### Official Review · AnonReviewer2 · 2018-11-03
**Contribution incremental and limited**

**Rating:** 4
**Confidence:** 4

**Review:**

The paper discusses a method for learning neural networks with hard-threshold activation. The classic perceptron network is a one-layer special case of this. This paper discusses a method called guided random local search for optimizing such hard-threshold networks. The work is based on a prior work by Friesen&Domingos (2018) on a discrete target propagation approach by separating the network into a series of Perceptron problem (not Perception problems as quoted by this paper).
I feel the proposed method is mainly formulated in Equation (3), which makes sense but not very surprising. The proposed random local search is not very exciting either. Finally, the empirical results presented do not seem to justify the superiority of the proposed method over existing methods. Overall, the paper is too preliminary.

---

### Official Review · AnonReviewer3 · 2018-11-07
**Good idea but far from a proper publication**

**Rating:** 3
**Confidence:** 5

**Review:**

TargetProp

This paper addresses the problem of training neural networks with the sign activation function. A recent method for training such non-differentiable networks is target propagation: starting at the last layer, a target (-1 or +1) is assigned to each neuron in the layer; then, for each neuron in the layer, a separate optimization problem is solved, where the weights into that neuron are updated to achieve the target value. This procedure is iterated until convergence, as is typical for regular networks. Within the target propagation algorithm, the target assignment problem asks: how do we assign the targets at layer i, given fixed targets and weights at layer i+1? The FTPROP algorithm solves this problem by simply using sign of the corresponding gradient. Alternatively, this paper attempts to assign targets by solving a combinatorial problem. The authors propose a stochastic local search method which leverages gradient information for initialization and improvement steps, but is essentially combinatorial. Experimentally, the proposed algorithm, GRLS, is sometimes competitive with the original FTPROP, and is substantially better than the pure gradient approximation method that uses the straight-through estimator.

Overall, I do like the paper and the general approach. However, I think the technical contribution is thin at the moment, and there is no dicussion or comparison with a number of methods from multiple papers. I look forward to discussing my concerns with the authors during the rebuttal period. However, I strongly believe that the authors should spend some time improving the method before submitting to the next major conference. I am confident they will have a strong paper if they do so.

Strengths:
- Clarity: a well-written paper, easy to read and clear w.r.t. the limitations of the proposed method.
- Approach: I really like the combinatorial angle on this problem, and strongly believe this is the way forward for discrete neural nets.

Weaknesses:
- Algorithm: GRLS, in its current form, is quite basic. The Stochastic Local Search (SLS) literature (e.g. [1]) is quite rich and deep. Your algorithm can be seen as a first try, but it is really far from being a powerful, reliable algorithm for your problem. I do appreciate your analysis of the assignment rule in FTPROP, and how it is a very reasonable one. However, a proper combinatorial method should do better given a sufficient amount of time.
- Related work: references [2-10] herein are all relevant to your work at different degrees. Overall, the FTPROP paper does not discuss or compare to any of these, which is really shocking. I urge the authors to implement some or all of these methods, and compare fairly against them. Even if your modified target assignment were to strictly improve over FTPROP, this would only be meaningful if the general target propagation procedure is actually better than [2-10] (or the most relevant subset).
- Scalability: I realize that this is a huge challenge, but it is important to address it or at least show potential techniques for speeding up the algorithm. Please refer to classical SLS work [1] or other papers and try to get some guidance for the next iteration of your paper.

Good luck!

[1] Hoos, Holger H., and Thomas Stützle. Stochastic local search: Foundations and applications. Elsevier, 2004.
[2] Stochastic local search for direct training of threshold networks
[3] Training Neural Nets with the Reactive Tabu Search
[4] Using random weights to train multilayer networks of hard-limiting units
[5] Can threshold networks be trained directly?
[6] The geometrical learning of binary neural networks
[7] An iterative method for training multilayer networks with threshold functions
[8] Backpropagation Learning for Systems with Discrete-Valued Functions
[9] Training Multilayer Networks with Discrete Activation Functions
[10] A Max-Sum algorithm for training discrete neural networks

---

> ### Author Response · Authors · 2018-11-14
> **Thank you for the detailed comments**
>
> Thank you for the detailed comments. We will certainly refer to [1] as we brainstorm approaches to address scalability and further improve our method, as advised by both you and Reviewer #1. Thank you also for the survey of literature on approaches different from FTPROP and GRLS for training neural networks with binary activation functions [2]-[10]. We were not aware of these works. Indeed a proper justification for target propagation methods in general (Friesen and Domingos, as well as ours) should compare against the state of the art in the area. Our group is just as surprised as you re that the FTPROP paper did not do this, and we will be trying to find the subset of [2]-[10] (and the broader literature) which represents leading alternative techniques to test against in a future iteration of this paper.

---

> > ### Comment · AnonReviewer3 · 2018-11-14
> > **Looking forward to the next version**
> >
> > Great, I wish you the best of luck!

---

### Meta-Review · Area_Chair1 · 2018-12-19
**Good idea, but research not yet ripe. Missing extensive comparison with alternative approaches.**

**Recommendation:** Reject
**Confidence:** 5

**Metareview:**

The paper proposes a novel  local combinatorial search algorithm for the discrete target propagation framework of Friesen & Domingos 2018, and shows a few promising empirical results.

Reviewers found the paper well written and clear, and two of them were enthusiastic about the direction of this research.
But all reviewers agreed that the paper is too preliminary, particularly in its empirical coverage. More extensive experiments are needed to compare with competitive approaches form the literature, for the task of training hard-threshold networks. Experiments would need to evaluate the algorithms on larger models and data more representative of the field, to measure how the approach can scale, and to convince of the superiority or advantage of the proposed method.